# Potential Cosmetic Active Ingredients Derived from Marine By-Products

**DOI:** 10.3390/md20120734

**Published:** 2022-11-24

**Authors:** Evi Amelia Siahaan, Ratih Pangestuti, Kyung-Hoon Shin, Se-Kwon Kim

**Affiliations:** 1Research Centre for Marine and Land Bioindustry, National Research and Innovation Agency (BRIN), Lombok Utara 83352, Indonesia; 2Research Centre for Food Technology and Processing, National Research and Innovation Agency (BRIN), Yogyakarta 55861, Indonesia; 3Department of Marine Sciences and Convergent Technology, Hanyang University, ERICA Campus, Ansan 15588, Republic of Korea

**Keywords:** marine, by-product, discards, skin health, cosmeceutical

## Abstract

The market demand for marine-based cosmetics has shown a tremendous growth rate in the last decade. Marine resources represent a promising source of novel bioactive compounds for new cosmetic ingredient development. However, concern about sustainability also becomes an issue that should be considered in developing cosmetic ingredients. The fisheries industry (e.g., fishing, farming, and processing) generates large amounts of leftovers containing valuable substances, which are potent sources of cosmeceutical ingredients. Several bioactive substances could be extracted from the marine by-product that can be utilized as a potent ingredient to develop cosmetics products. Those bioactive substances (e.g., collagen from fish waste and chitin from crustacean waste) could be utilized as anti-photoaging, anti-wrinkle, skin barrier, and hair care products. From this perspective, this review aims to approach the potential active ingredients derived from marine by-products for cosmetics and discuss the possible activity of those active ingredients in promoting human beauty. In addition, this review also covers the prospect and challenge of using marine by-products toward the emerging concept of sustainable blue cosmetics.

## 1. Introduction

According to the recorded statistics from the Food and Agriculture Organization (FAO), total world fisheries and aquaculture production in 2020 reached 240 million tons. It is estimated that the fisheries production number will remarkably increase every year because of the growing demand for marine products [1]. The current global production growth of marine resources encouraged economic strength and increased food supply capacity. However, a massive number of marine productions create a considerable amount of marine discard that causes negative impacts (e.g., environmental pollution and economic loss). According to the FAO, the annual discards from world fisheries were approximately 20 million tons and composed of processing leftovers; by-products (fins, heads, skin, guts, scales, shells, and bones); and unwanted species [2,3]. Regarding this important issue, strategic utilization of marine discards/by-products is needed to not only valorize waste materials but also provide environmental benefits. Moreover, utilizing marine discards/by-products could encourage a zero-waste strategy and pursue the Sustainable Developmental Goals (SDGs) of the United Nations, particularly SDG 14, to conserve and sustainably use the oceans, seas, and marine resources for sustainable development [4].

Developing valuable products from marine discards/by-products is not new. To date, most of the marine discards/by-products are traditionally utilized as low-commercial products (e.g., silage, fishmeal, fertilizer, fishing bait, and raw aquafeed) [5,6,7,8]. In contrast, they could be better utilized as high-value products, including the production of novel cosmeceutical ingredients. There has been growing interest in using marine discards/by-products for personal care products because marine discards/by-products contain the same valuable active compounds as the marine organism itself. At present, few studies have been reported on the production of cosmeceutical active ingredients from marine discards/by-products. This review focuses on cosmeceutical active compounds from marine discards/by-products and their potential application in the cosmeceutical industry.

## 2. Potential Cosmeceutical Ingredient from Marine By-Products

By-products are valued raw materials with great prospects for various valuable products (Figure 1). Marine discards, which contain meat and viscera, have been limitedly exploited for fish meals, fish oil, and silages [9]. Due to their unique functionality and marked biological activities, marine by-products have been suggested to be valorized as raw materials of fine chemicals (e.g., pharmaceuticals and cosmeceuticals), and those will provide a significant additional income for companies that use marine by-products [10,11]. This part reviews the potential bioactive molecules of marine by-products as raw materials for cosmetic ingredients that are recovered from leftover seafood processing of finfish, crustaceans, shellfish, and other marine commodities.

### 2.1. Cosmetic Active Ingredients from Finfish By-Product

The by-products of the finfish processing plan are usually composed of heads (9–12% of total fish weight percentage), intestine (12–18%), skin (1–3%), bones (9–15%), and scales (~5%), where its composition might vary and be reliant on size, species, category process, and technology used [12]. The fish skins, bones, and scales are labeled as finfish by-products containing valuable materials for cosmeceutical application (e.g., collagen, gelatin, fish oil, and calcium phosphates).

#### 2.1.1. Collagen, Gelatin, and Collagen Derivatives as Cosmetic Ingredients

Collagen is beneficial as an active beauty ingredient with good biocompatibility and minimal immunogenicity on the human body. Therefore, it is in great demand as an ingredient in cosmetic products. Generally, collagen type I, including collagen recovered from fish scale, bone, scale, and swim bladder, is more desirable for cosmetical as human skin is predominantly by the same type of collagen, so it is compatible with human skin [13]. Marine collagen constitutes a crucial alternative to porcine and bovine collagen since land base commercial sources have been associated with zoonotic diseases (e.g., transmissible spongiform encephalopathy, bovine spongiform encephalopathy, and foot and mouth disease) [14]. In addition, consumer concerns about halal cosmetics have risen recently in the Muslim community, where the gelatin ingredient and bioactive from porcine are prohibited and some from bovine allow for specific requirements [15]. Those showed opportunities for marine by-products-based collagen.

Fishbone, skin, scale, and swim bladders are rich in a collagen matrix. Collagen is a set of extracellular matrix structural proteins organized in a fibrillar arrangement [16]. Collagens are frequently structured by polypeptide chains (trihelix) made of repeated triplets of glycine and two other amino acids, the most frequent of which are proline and hydroxyproline (Hyp), out of a total of approximately 1000 amino acids [17]. Type I collagen is predominantly recovered from skin, tendon, bone, and scale, while type II collagen could be gained from cartilage fish [18]. Fish skin, bones, and scales contain collagen, which has a lower denaturation, melting temperature, and varied composition than mammalian collagen [19].

Marine collagen peptide or marine hydrolyzed collagen refers to the derivative product of collagen, which is hydrolyzed to produce a shorter peptide. Unlike native collagen, it usually has a molecular weight (MW) of 300–6000 Dalton (Da) formed in the free structure of peptide [20]. Small peptides and short polypeptides produced from collagen are conveniently included in cosmetic formulations, because they are easy to dissolve in water [21]. Collagen from mixed fish parts was treated with endoprotease followed by filtration using membrane-generated five fractions, bigger 30 kilos (k)Da, 10−30 kDa, 5−10 kDa, 1−5 kDa, and lower 1 kDa collagen peptide [22]. Hydrolyzed collagen of snapper fish scales was produced using protease flavorzyme [23]. The hydrolyzed collagens from the skin of a blue shark, a swordfish, and a yellowfin tuna were obtained by hydrolyzing using food-grade alsace from pepsin soluble collagen (PSC) [24]. The hydrolysate fish collagen can also produce using high temperature and pressure (210 °C and 2100 kPa) directly from tuna skin [25].

Several studies have demonstrated the potency of collagens from marine by-products for cosmetics ingredients. Topical exposure of collagen to skin showed that fish skin cod collagen had good moisturizing effects by absorbing water and avoiding skin dehydration with no signs of irritation on the skin [14]. According to a study, bovine collagen is a more effective cosmetic ingredient than gelatin or collagen hydrolysate because it promotes significantly higher rates of adhesion and proliferation in keratinocytes. [26]. The effectivity of fish collagen on skin health might pose different results since the human skin has different characteristics and the human body has limitations in absorbing high molecule peptides although that finding might work the same on marine collagen.

Collagen from marine by-products has not only a potential benefit through the topical application but also shows activity on skin beauty by oral administration. Fish skin collagen hydrolysate of demersal fish *Pollachius virens*, *Hippoglossus hippoglossus*, and *Pleuronectes platessa* significantly improved elasticity and sebum after two months of administration [27]. That product is marketed with the label CELERGEN [28]. After eight weeks of oral administration, the clinical testing on commercial fish collagen peptides (Peptan^®^F) showed that it significantly improved skin moisture and increased collagen density dermis [29]. Collagen peptides of fish cartilage (marketed as Cartidyss^®^) were supplemented orally for 90 to achieve clinical benefits for the skin [30]. Marine collagen peptides from chum salmon skin taken orally showed to speed up the healing of cutaneous wounds in rats [31].

Gelatin and its hydrolysates, derivatives of collagen peptides, are also explored for cosmetics, and they open various applications of collagen products to some extent [32]. Gelatin is obtained through partial hydrolysis of collagen using acid (type A gelatin)/alkali base (type B gelatin) followed by heat [33,34]. Thus, heat and acid/alkali turn insoluble protein collagen into water-soluble gelatin [35]. Since gelatin is partially hydrolyzed collagen, it usually has a lower MW of approximately 2–20 kDa, and its structure combines single or multistranded peptides, each with extended left-handed helix conformations and 50–1000 amino acids [36]. Gelatins have been derived from many marine by-products, namely shark cartilages [37], swim bladder of tuna [38], yellowfin tuna skin [39], and bone of red snapper and grouper [40]. Generally, fish gelatin had lower melting temperatures than mammal gelatin (32 °C–35 °C) [41].

The biological properties of the gelatin could be improved by producing gelatin peptides, and enzymatic is an efficient way to do so [42]. Alcalase, flavorzyme, neutrase, and protamex were used to hydrolyze skate fish (*Okamejei kenojei*) skin gelatin, and the resulting products had substantial antioxidants [43]. The potential active antioxidant hydrolysate was obtained by hydrolyzing the thornback rays (*Raja clavata*) skin gelatin using proteases from Bacillus subtilis [44]. Furthermore, the skin and scale gelatin of barred mackerel was enzymatically hydrolyzed by alcalase followed by actinidin [45]. Gelatin and hydrolyzed peptides are widely used in cosmetics as they have gel- and film-forming abilities, act as thickening agents and show bioactivity to promote skin health [46]. Numerous cosmetic items containing gelatin have been developed (e.g., face creams, body lotions, shampoos, hair sprays, sunscreens, bath salts, and bubble baths) [47].

The above scientific pieces of evidence show the effectiveness of collagen, gelatin, and their derivatives from marine by-products as active ingredients in cosmetics. It is essential to utilize collagen and collagen derivatives from marine by-products as ingredients in personal care products because of religious restrictions, where porcine gelatin and bovine gelatin are the primary sources of cosmetic ingredients. Halal or cosher gelatin is a critical ingredient in the development of halal and cosher beauty products, where marine by-products gelatin is the answer to that issue.

#### 2.1.2. Active Ingredients of Fish Oil for Cosmetic

Fish oil is a cosmetic ingredient that could be recovered from marine by-products. Marine fish is recognized as a good source of fish oils. The fish oil by-product can be obtained from the head, frame, trimming, viscera, and skin of salmon by-fillet processing [48] and tuna processing [49], and the by-products of oily fish canings [12]. The IFFO, a marine ingredient organization, reported that 51% of world fish oil is recovered from seafood processing discard [9].

The active properties of fish oil to promote skin health are mainly related to polyunsaturated fatty acids content, such as docosahexanoic acid (DHA) and eicosapentanoic acid (EPA). Benefits from the fish oils could be obtained by oral administration as a supplement. A comprehensive review of the role of fish oil and its constituent in cosmetics has been reviewed by Huang et al. [50]. A clinical trial showed that dietary lipids, including omega-3 fatty acids, could have meaningful impacts on skin reaction to ultraviolet (UV) in a relatively short time and at low doses [51]. That finding supported a study on animals receiving menhaden oil (rich in omega-3); it required twice the irradiance level to induce equivalent erythema in corn oil-fed [52]. Supplementing with fish oil is related to a decrease in the acuteness of acne overall, especially in people with moderate to severe acne [53]. Furthermore, panelists showed decreasing UVB irradiation-induced erythema and induction of p53 post three months of oral administration of 4 g/day EPA [54]. In addition, an in vivo study showed that the liver and skin oil of the Antarctic fish (marbled rock cod) could be used as bioactive nutrients to improve skin health since they significantly suppressed matrix metalloproteinase-1 (MMP-1) formation [55]. A clinical trial was also conducted on commercialized fish oils to investigate their benefit on human skin. Itching, erythema, and scaling in patients with persistent chronic psoriasis significantly decreased after eight weeks of treatment with ten fish oil capsules (marketed as MaxEPA) [56].

#### 2.1.3. Natural Calcium Phosphates (CaPs) as Cosmetic Ingredients

The natural calcium phosphates (CaPs), a group of substances and minerals, form the inorganic component of the bone and teeth of vertebrates. Currently, CaPs are mainly applicated in biomaterials medicine for rejuvenating or replacing bone tissue [57]. The CaPs can be synthetically produced or recovered from biogenic sources [58]. Fish scales and bones have recently been studied as a source of natural calcium phosphates [59]. Chen et al. [60] calcinated sole fish bone at 700 °C–800 °C to obtain calcium phosphate, mainly containing β-tricalcium phosphate (β-TCP). Hydroxyapatite (HA) was recovered from the tuna scale using an alternate immersion aqueous solution followed by several steps of calcined at 250 °C–550 °C [61]. Silver containing HA and biphasic HA/β-Tricalcium Phosphate (TCP) were obtained by sintered cod fish bone at 1250 °C [62]. Hamada et al. [63] recovered HA from the bone of horse mackerel by ashing at 600 °C and found the trace elements as minor constituents. The biphasic calcium phosphates (HA-TCP) were produced by sintering salmon bone at 1000 °C–1300 °C [64].

Synthetic and natural CaPs are broadly applied for cosmetic goods, such as skin care, hair care, deodorant care, and oral care [65], however, a limited number of studies on fish-derived CaPs as cosmetic ingredients. Piccirillo et al. [66] and Teixeira et al. [67] investigated the potency of HA from the cod fish bone. The HA-Fe_2_O_3_ powder presented effective absorption properties in the full-range UV and did not pose radicals when irradiated. The sun cream prepared from that substance exhibited high UVA and UVB absorption and can be utilized as a broad-spectrum sunscreen. Skin hydrogels, which contain unmodified HA, Mn-doped (HA-Mn), and Fe-doped (HA-Fe), were prepared from Fringescale sardinella bones. The hydrogels showed different sunscreen protection factors: SPF 20, SPF 40, and SPF 50 for HA, HA-Mn, and Ha-Fe, respectively [68]. Future study of marine-by-product CaPs is needed to provide sufficient scientific evidence for cosmetics application because the marine-derived CaPs might have different trace minerals content with CaPs-based terrestrial animals, and it could provide different range activity.

### 2.2. Cosmetic Active Ingredients from Crustacean By-Products

The crustacean seafood processing, mainly prawns, shrimps, crabs, and lobsters, generates considerable leftovers, consisting of the carapace and head. The seafood processing of shrimp, crab, lobster, crayfish, and krill leftover is approximately 60–80%, 60–70%, up to 60%, up to 85%, and 70–75%, respectively [69]. That by-product might be fractioned into three primary chemicals, which are 20–50% calcium carbonate, 15–40% chitin, and 20–40% protein [70]. Several cosmetic ingredients and actives could be recovered from crustacean by-products, such as chitin, chitosan, and astaxanthin.

#### 2.2.1. Chitin and Its Derivatives as Cosmetic Ingredients

Chitin (C_8_H_13_O_5_N)_n_ is the second most abundant carbohydrate after cellulose. It is a highly hydrophobic, odorless, and tasteless compound composed of a linear poly-β-(1,4)-N-acetyl-D-glucosamine and is white or yellowish [71]. Chitin and cellulose have a similar structure, but chitin owns an acetamide group (NHCOCH_3_) at the C2 position, which plays an essential role in influencing characteristics and turning it into a versatile chemical [71]. Crab and shrimp shells have been recognized as commercial sources of chitin and chitosan [72]. Chitosan is derived from chitin by additional alkaline treatment, namely deacetylation. It comprises a random copolymer of a d-glucosamine and N-acetyl-(1-4)-d-glucosamine. Chitosan differs from chitin based on the percentage of acetylation (DA%), whereas chitosan defines as lower than 50% of DA [73,74]. Furthermore, chitosan has a distinct ability to become polycationic when dissolved in an acidic solution (pKa = 6.0) and produces a viscous solution [75]. Chitosan is more applicable than chitin, and its applications medical and pharmaceutical are constrained by its viscosity and MW [76]. Furthermore, chitosan can be converted into chitooligosaccharide or chitosan oligomer, which has a different deacetylation and polymerization degree to lower its MW. Chitosan oligomer can be applied in food, cosmetic, biomedical, and agricultural applications [77,78].

Chitin and its derivatives, with their distinct biological and technological distinctive, are of concern cosmetic and cosmeceutical. They could be used as a delivery system and active ingredients in cosmetics. Aranaz et al. [79] reported that European Commission compiled as many as 44 cosmetic substances as chitosan and derivates and eight as chitin and derivatives.

Chitosan and its derivatives could work as a delivery system of active compounds, and they pose synergy with active compounds generating optimal benefits for human skin health. As the delivery system for topical application, chitosan is sometimes formed to be smaller or typically shaped. The chitosan nanoparticles have been used as an active carrier and stabilizer for encapsulating natural and purified annatto and saffron (sunscreen agents) to prepare UV-protective cosmetic emulsions. Those emulsions showed good storage and color stability for up to ninety days at room temperature. Despite exhibiting low cytotoxicity, the emulsion formulas displayed SPF values of 2.15–4.85 [80]. Furthermore, a combination of the PBSA and chitosan microparticle enhanced the UV screening effect as showed by in vitro test of chitosan gel and Transpore^TM^ tape test. The origin chitosan gel had an in vitro SPV value of 0.89, and chitosan microparticle-PBSA could generate SPF values up to 1.61 [81].

The physical properties of chitin and chitosan (e.g., film-forming ability and compatibility with other materials) open more options for applications as film and emulsion for beauty purposes. Morganti et al. [82] demonstrated the high compatibility of chitin and chitosan nanofibril-hyaluronan as carriers, both as nanoemulsions and nonwoven films, to distribute antiaging compounds through the skin. The result showed that chitosan-hyaluronan completely released lutein in 20 h, while the chitin-hyaluronan took a longer time, almost double. The in vitro test of nanoemulsion containing active liposomal complex showed increased skin collagen growth, while the nanoemulsion enriched with entrapping the antioxidant complex melatonin-vit E-betaglucan (MEB) inhibited approximately 90% of collagenase. The in vitro assay, including oral and topical administration of chitin nanofibril-hyaluronan MEB, expressed that the formulation promoted repairing the skin as affected by photo irradiation, acne, and rebalancing hydration of the dermal. Furthermore, Morganti et al. [83] suggested using complex chitosan nanofibril and nanolignin as an eco-friendly beauty mask to deliver the active ingredient. The antiaging activity of beauty masks repressed MMP-1, elevated type I collagen production, and suppressed skin aging. The preventive and rejuvenating functions demonstrated by an in vitro study conducted on 30 voluntarily participating women displaying signs of photoaging have been verified by the early in vivo results achieved through engineering approaches. This combination generated an alternative for eco-friendly beauty mask.

In addition, chitosan film exhibited a good biodegradable cosmetic mask with good flexibility, retaining water, and compatibility with active ingredients. The antiaging ingredients, namely annatto powder and vitamin C, were trapped in reacetylated chitosan matrix films resulting in a more flexible and selective permeability film than pure chitosan matrix films. The reacetylated chitosan film also released active ingredients faster, where approximately 52% of the active compounds were released in 15 min. It did not pose cytotoxic, as well as intracellular, oxidation [84]. Furthermore, the chitosan–alginate nanoparticle also showed antibacterial activity against acne bacteria, Propionibacterium acnes, as it distributed the membrane cell of bacteria. Chitosan–alginate also demonstrated anti-inflammatory effects by preventing human monocytes and keratinocytes from producing inflammatory cytokines in response to P. acnes infection [85].

The oligomer chitosan possessed an antibacterial effect on Propionibacterium acnes [86]. Oligochitosan (MW = 10 kDa) presented a high antimicrobial effect on Propionibacterium acnes (minimum inhibitory concentrations; MICs = 32–64 μg/mL) [86]. That oligomer incorporated with tetracycline and erythromycin (model antibiotics) also showed excellent inhibition on P. acnes, resulting in a median fractional inhibitory concentration of 0.02–0.56. Therefore, the oligochitosan–antibiotics complex was suggested for treating antibiotic-resistant bacteria [86]. In addition, carboxymethyl chitosan has been suggested as a cosmetic ingredient, such as the delivery system, moisture-retention agent, antimicrobial, antioxidant, and naturally derived emulsion stabilizer [87].

Research about chitin and chitin derivatives for cosmetics mainly focuses on topical application. Further studies on the oral administration of chitin and chitin derivatives on skin health are needed to better understand the cosmeceutical properties of chitin and chitin derivatives.

#### 2.2.2. Astaxanthin as Cosmetic Ingredients

The by-product of crustacean food processing is a significant source of natural astaxanthin. Astaxanthin (3,3′-dihydroxy-β, β′-carotene-4,4′-dione) is a spinoff of xanthophyll carotenoid, which is a red fat-soluble pigment. In crustaceans, astaxanthin is established and complexed into a protein known as carotenoprotein or in free form, and its amount in shellfish differs on species and season [88]. Astaxanthin, which is connected to free protein or chitin, is found in significant amounts in the carcass of crustaceans ranging from 2.9 to 1203 μg/g of dry matter [89]. Astaxanthin is the most important carotenoid in crustacean shells [90]; however, microalgae-derived astaxanthin is commonly used in cosmetics [91].

Astaxanthin could be recovered from crustacean carapaces through numerous methods, which are organic solvent, oil, and the green extraction technique [92,93]. The astaxanthin was extracted from dried shells of red swamp crayfish using ethanol, followed by column chromatography to obtain pure astaxanthin [94]. Ahmadkelayeh et al. [95] extracted astaxanthin using fish oil as an alternative vegetable oil and a process acquired an optimal yield at 65 °C, biomass:oil ratio of 9:1 for 1.5 h. A higher astaxanthin yield was obtained from shrimp carapace by pre-fermented with Saccharomyces cerevisiae, followed by hexane extraction with acetone at a ratio of 1:1 [96]. Furthermore, it also could be isolated from crab shells by a sequence of supercritical fluid and microwave extraction [97]. In addition, astaxanthin also could be extracted by an integrated biorefinery model with astaxanthin, protein, and chitin as targeted products [98].

The topical application and oral administration of astaxanthin show short skin health advantages. It could act as ray protection, antioxidant, antiaging, anti-wrinkle, moisturizer, wound healing, anticancer, and anti-eczema. In vitro assay showed that astaxanthin act as an antioxidant and promotes skin cell development. Instead of directly scavenging free radicals, the antioxidant activity of astaxanthin may work by encouraging the production of modest levels of reactive oxygen species (ROS) in cells [99]. Astaxanthin extract affected human dermal fibroblast proliferation via VEGF upregulation. Enriched astaxanthin extract increased tissue inhibitor of metalloproteinase-1 (TIMP1), which, in turn, decreased MMP proteins and increased collagen levels, showing the benefit of astaxanthin extract to activate human skin cell development [100].

Despite oral administration, the benefit of astaxanthin could be obtained through the topical application, as shown by an in vivo assay. Skin thickening caused by UV exposure could be avoided by applying astaxanthin–liposomal to the skin beforehand. Pre-administration of astaxanthin–liposomal also reduced collagen loss brought on by UV exposure [101]. Additionally, topical application of astaxanthin–liposomal cationic lipid suppressed melanin synthesis in UV-exposed skin. An in vivo assay showed that topical astaxanthin–liposomal stopped UV-induced skin damage [101]. In addition, egg phosphatidylcholine–liposomes encapsulated carotene or tocopherol were less effective than astaxanthin-containing liposomes, and the cytotoxicity test showed that astaxanthin–egg phosphatidylcholine–liposomes could protect a dermal fibroblast cell of a mouse (NIH3T3 cells) [102].

Astaxanthin also could be delivered as the emulsion and film for skin beauty purposes. Nanoemulation of astaxanthin–carboxymethyl chitosan produced by the low-energy emulsion phase showed good permeability in the skin and better stability with no toxicity effect on skin cells [103]. In addition, in vivo assay showed squid skin astaxanthin–collagen film treatment promoted increasing collagen content, fibroblasts, granulation, scar thickness, effective neovascularization, and quicker epithelialization of excisional and incisional Wistar rats in a short period [104].

There are fewer reports regarding the astaxanthin of marine by-products even though there are scientific pieces of evidence regarding the benefit of astaxanthin for cosmetic products. More clinical research on the cosmeceutical benefits of astaxanthin from marine by-products is needed, since animal-based astaxanthin may pose different responses to human skin health.

### 2.3. Cosmetic Active Ingredients from Molluscan By-Products

In 2022, mollusks and other aquatic invertebrates shared approximately 11% of international trade seafood commodities, where scallops, clams, oysters, and mussels are major bivalve mollusk species for international trade [1]. Those commodities also generate a large amount of shell waste which makes up approximately 90% total mass of mollusks [105]. In addition, the shell contains compounds promoting skin health and beauty.

Generally, the mollusk shell typically comprises three layers: an organic periostracum on the outside, a calcified prismatic layer in the middle, and a calcareous nacreous layer on the inside. Conchiolins, which are mainly insoluble proteins, make up the periostracum. Conchiolin, calcite prisms, and aragonite tablets are present in the nacreous and prismatic strata [106]. Furthermore, the nacre, the mother of the pearl, is a calcified structure that creates the inner layer in bivalve shells. It comprises aragonite (approximately 95–97%) tablets oriented in multiple layers, each bounded by an organic matrix. That matrix, which shares approximately 5% of the nacre structure, is primarily made up of proteins and polysaccharides [107]. The powdered pearl shells or powdered nacreous shell layer has become an ingredient of interest for cosmetics [108].

The water-soluble nacre isolated from the fine powder of pearl oysters (Pteria martensii) promoted wound healing in the deep burned skin of porcine [109]. The topical (spraying) application of water-soluble nacre directly on the burned skin promoted collagen formation, restoring the skin to a normal state. Those in vivo findings supported the in vitro assay on water-soluble nacre treated fibroblast NIH3T3 cells, where the water-soluble nacre treated increased proliferation and collagen synthesis [109]. Almost similarly, the whole water-soluble matric and fraction (SE4) of Pinctada maxima promoted the differentiation of cells in vitro [110]. The nacre of P. maxima, which contains mainly carbonate and 17% fibrous protein, enhanced collagen synthesis in a rat skin incisional injury model [111].

Furthermore, Agarwal et al. [108] found powdered nacre of *P. margaritifera* shell having limited cytotoxicity at high concentrations and exhibiting no apparent oxidative stress on primary skin fibroblast cells and epidermal skin cells. The hydrogel-trapped nacre pearl powder was investigated for anti-inflammation and anti-apoptosis. The result depicted that nacre did not show toxicity on HaCaT cells, and it reduced the inflammation and apoptosis of HaCaT cells [112]. Rousseau et al. [113] elucidated and tested the lipid constituent of nacre. That lipid, which contains 79.03% squalene-like lipid, stimulated a reconstitution of the intercellular content of the stratum corneum on atopic dermatitis skin. Furthermore, the protein conchiolin, which has 17 amino acids, may be responsible for the positive effects of pearl powder on wound healing [114].

The extract of scallop and mussel shells also has potential use for dermo–cosmetic applications. In an in vivo assay, the acid extract of scallop shell fractions induced types I and III collagen synthesis [115]. In addition, the scallop shell extract posed the activity to promote collagen metabolism in skin fibroblast cells in vitro. Then, the topical application of the scallop shell extract on the rat dorsal skin improved the collagen content of the skin tissue segment [116].

### 2.4. Cosmetic Active Ingredients from Seaweed Biomass Waste

Seaweed processing usually targets major components, such as agar, carrageenan, and alginate, and the rest of the material, as minor components, might be discarded or lost during processing, recognized as impurities. Some minor components have been recognized as potential cosmetic ingredients, such as phycocyanin, phycoerythrin, and mycosporine-like amino acids (MAAs). Therefore, integrated biorefinery concepts have been proposed in seaweed process lines with hydrocolloids and other active compounds as targeted products [117]. The high-economic seaweed species contain various amounts of proteins depending on the species, and those kinds of proteins also have certain cosmetic functionality. Phycobiliproteins and MAAs are functionally active proteins widely recognized for cosmetics [118]. Both active compounds could be found in economic red algae (agarophyte and caragenophyte) [119].

The phycobiliproteins are an essential group of precious macroalgal compounds [120]. Phycobiliproteins (phycocyanin and phycoerythrin) are natural cosmetics colorants despite having less heat and light stability [121]. These natural dyes have been applied to several daily cosmetic products, such as lipsticks or eyeliners [122,123]. The phycobiliproteins act as the colorant and possess antioxidant and anti-inflammatory [124,125].

The MAAs are crucial for sunlight absorption and protects marine organisms from UV radiation [126]. The MAAs content in seaweed varies between species and seasons [127], and red algae have been suggested as the potential source of MAAs [128]. Ryu et al. [129] showed that the methanol extract of Corallina pilulifera methanol had vigorous antioxidant activity and protective effects against UVA-induced oxidative on human skin fibroblast. MAAs-Porphyra-334 from Porphyra protected human skin fibroblasts from UVA-induced photoaging through suppressed ROS production and the expression of MMPs, as well as increasing the extracellular matrix component levels of procollagen, type I collagen, and elastin [130].

Liquor waste generated from the post-food processing of brown algae Hijiki, Sargassum fusiforme, was found to have high antioxidant capacity and tyrosinase (TYR) inhibition. Hijiki liquor waste acted as a potent TYR inhibitor with an IC_50_ of 3.1 µg/mL after being fractionated with class column HP_2_O followed by methanol (MeOH). The MeOH fractions of this algae waste could also inhibit melanin production on a 3D human skin model with topical application. No cytotoxicity was observed when 20 mg/mL of MeOH fractionation was applied to the tissue [131].

## 3. Cosmeceutical Properties of Compounds from Marine By-Products on Skin Health

### 3.1. Skin-Whitening Properties

Recently, the market for skin-whitening products has grown tremendously in the Asia-Pacific region, driven by the desire to have a brighter and lighter skin tone. In Asian culture, lighter skin tone has long been associated with youth, beauty, and prosperity. This perspective drives many Asian women to become obsessed with having a flawless and lighter skin tone and affects the raising demand for skin-whitening products. Human skin coloration has a wide range of tones and colors that are influenced by intrinsic and external factors, including the type and amount of melanin in the skin, genetics, the number of melanosomes, UV exposure, and environmental pollution [132,133]. Melanin is a major pigment that is produced by melanocytes in the epidermis through a process of melanogenesis. It is the most prominent factor in determining skin color, which can be classified into two types: eumelanin (black to brown pigment) and pheomelanin (red to yellow pigment). It plays an important role in protecting the skin against the harmful effects of UV radiation (UVR) and oxidative stress. It has also been reported that melanin could prevent the development of skin cancer by shielding cells from UV-induced DNA damage and killing [134,135]. Despite its benefits on skin health, the excessive production of melanin results in serious skin problems, including freckles, melasma, solar lentigines, and pigmentation [133,136]. Hence, the regulation of melanogenesis is important for controlling skin pigmentation.

In the melanogenesis pathway, three main melanogenic enzymes are involved, including TYR and tyrosinase-related proteins 1 and 2 (TYRP-1 and TYRP-2). Tyrosinase, a key regulatory enzyme, initiates the rate-limiting step by oxidating the L-tyrosine to L-3,4- dihydroxyphenylalanine (L-DOPA), and L-DOPA to o-dopaquinone (DQ). Following the DQ formation, TYRP-2 leads the conversion of dopachrome to 5,6-dihydroxindole (DHI) or indole 5,6-quinone 2-carboxylic acid (DHICA) and, finally, results in eumelanin production. In the presence of cysteine, the DQ is converted into cysteinyl DOPA, which is then oxidized to produce pheomelanin [136,137]. TYR plays the dominant role in melanogenesis, and suppressing its activity could control melanogenesis. TYR becomes the main cellular target for skin pigmentation treatment [138]. Therefore, numerous compounds of TYR inhibitors have been developed lately.

In the search for new TYR inhibitors, it was found that functional compounds from marine by-products/discards showed potent inhibitory activities against TYR. Protein hydrolysate of shrimp by-product *Metapenaeus monoceros* has shown profound TYR inhibition activity with an IC_50_ of 6.13 µg/mL. The protein hydrolysate of shrimp by-product at 400 µg/mL concentration displayed TYR inhibition of 100% [139,140]. Chintong et al. [141] proved that astaxanthin from shrimp shells *Litopenaeus vanamei* has significantly inhibited TYR in a dose-dependent manner at concentrations of 3–50 µg/mL. The TYR inhibition of astaxanthin from shrimp waste may be attributed to the presence of two oxygenated groups, which can chelate the copper of TYR [142]. Similarly, the tilapia scale polypeptide possesses the copper chelating ability of TYR. Further, the in vitro studies have shown that the polypeptide hydrolysate of tilapia by-product could strongly suppress TYR activity at a concentration of 5 µg/mL and effectively reduce the melanin production in the mouse melanoma cells (B16-F10) [143]. Apart from being used for hyperpigmentation treatment of human skin, as well as animals, TYR inhibitors isolated from marine by-products have suggested a great potential to cosmeceutical industries because of their skin-whitening effect and depigmentation after sunburn. The photoprotective mechanism of cosmetic active ingredients from marine by-products on humans is shown in Figure 2.

Collagen hydrolysates from marine by-products act as a TYR inhibitor, as shown by in vivo and in vitro assays, as well as clinical tests [144,145]. Several in vivo assays, in vitro and clinical tests on active collagen peptides from fish by-products for topical applications have been reported. A fraction (DLGFLARGF, 498.2695 *m/z*; mas 994.5236) of hydrolyzed fish scale collagen showed a tyrosinase inhibition ability (IC_50_ = 3.09 mM). In addition, Val, Ala, Leu, and Ile are predicted to act as an inhibitor of dopaquinone formation, hence inhibiting melanin production [144]. Milkfish scale collagen peptide could moisturize, prevent antiaging, and whiten the skin, as it showed an excellent capacity of moisture absorption (20%), inhibited tyrosinase activity (IC_50_ = 752.4 µg/mL), and melanin production (IC_50_ = 887.1 µg/mL) [145]. The clinical trial test showed that the serum enriched in marine collagen peptide compounds showed a moisturizing effect in short-term applications [146]. The 5–10 kDa fraction of tuna skin collagen peptide showed antiaging (inhibiting tyrosinase and gelatinase), and the <1 kDa fraction had antioxidant activity and had been suggested for cosmetic purposes [25].

### 3.2. Antiaging and Skin Rejuvenation Properties

Marine products (e.g., marine processing by-products) have been the subject of intensive investigation and are reported to be potential antioxidant, antiaging, and skin rejuvenation properties. This combination of biological properties makes marine processing by-products a unique skin care candidate with antiaging and skin rejuvenation properties.

Marine processing by-products, such as chitin, chitosan, chitooligosaccharides (COS), collagen, gelatin, and bioactive peptides, have been reported to possess potent antioxidant activity. The antioxidant activity of these marine processing by-products has been determined by various methods of antioxidant assays, such as 1,1-diphenyl-2-picryl hydrazil (DPPH) radical scavenging, lipid peroxide inhibition, ferric-reducing antioxidant power (FRAP), β-carotene bleaching methods, nitric oxide (NO) scavenging, 2,2′-azino-bis-3-ethylbenzothiazoline6-sulfonic acid radicals (ABTS) radical scavenging, 2-thiobarbituric acid, superoxide anion radical and hydroxyl radical scavenging assays (Table 1) [147,148].

The antioxidant properties of collagen, gelatin, and bioactive peptides derived from marine processing by-products are strongly related to their MW, amino acid types, and sequences. Commonly, collagen, gelatin, and bioactive peptides with lower MW will show stronger antioxidant effects. Collagen with lower MW has a greater ability to donate an electron or hydrogen to stabilize free radicals. In addition, Wang et al. (2013) reported that the antioxidant activity of collagen is also related to the presence of hydrophobic amino acid residues within the sequences [149].

Chitollogisaccharides (COS), the oligosaccharide form of chitosan, have been demonstrated to possess antiphotoaging properties [150,151,152]. The topical application of COS in mice skin for 10 weeks has been demonstrated to alleviate the macroscopic and histopathological damage of mice skin. COS treatment also modulated proinflammatory mediators and upregulated antioxidant enzymes [152]. The COS with MW 3–5 kDa has been demonstrated to inhibit the expressions of MMP-1, MMP-8, MMP-9, and MMP-2. The photoprotective effects of COS in UVA-irradiated human dermal fibroblast were found to be mediated by the activator protein-1 (AP-1) signaling pathway [151]. Afonso et al. [84] prepared chitosan and reacetylated chitosan films added with annatto seed powder and vitamin C for prospective use as a cosmetic mask. The reacetylation chitosan displayed greater water affinity, an amorphous microstructure, and more elastic and less resistant films than the native chitosan. All films were found to be noncytotoxic, as tested in human keratinocytes (HaCaT cells), and suppressed the intracellular oxidation process. In addition, the chitin nanoparticle has also been used to deliver antiaging active ingredients through the skin [82]. Zhao et al. [153] demonstrated that chitosan hydrogel-encapsulated extracellular vesicles were able to slow down the aging of the skin by improving the function of aging dermal fibroblasts. Chitosan hydrogels not only restricted to those activities but also promoted dermal fibroblast cell proliferation; downregulated the MMP-1, MMP-2, MMP-3 and MMP-9 expressions; and prolonged the release of extracellular vehicles. The hydrogel-based chitosan–fucoidan loaded with silibinin has also been demonstrated to possess a UVB photoprotective effect in hairless mice [154].

**Table 1 marinedrugs-20-00734-t001:** Bioactive peptides and hydrolysates from marine processing by-products with potential antioxidant activity.

Source	Enzymatic Hydrolysis	Organ	Sequence	Assay	Activity	Size	Ref.
Alaska pollack (*Gadus chalcogrammus*)	Alcalase, Pronase E, and collagenase	Skin	-	TBA, in vitro	-	-	[155]
Hoki (*Johnius belengerii*)	Trypsin	Skin	His-Gly-Pro-Leu-Gly-Pro-Leu	DPPH, carbon-centered, superoxide radicals, linoleic acid peroxide	-	797.00 Da	[148]
Croceine croaker (*Pseudosciaena crocea*)	Pepsin and Trypsin	Skin	Gly-Phe-Arg-Gly-Thr-Ile-Gly-Leu-Val-Gly	DPPH	IC_50_: 1.271 mg/mL	976.55 Da	[149]
				Superoxide radical	IC_50_: 0.463 mg/mL		
				ABTS radical	IC_50_: 0.421 mg/mL		
			Gly-Pro-Ala-Gly-Pro-Ala-Gly	DPPH	IC_50_: 0.675 mg/mL	526.24 Da	[149]
				Superoxide radical	IC_50_: 0.099 mg/mL		
				ABTS radical	IC_50_: 0.309 mg/mL		
			Gly-Phe-Pro-Ser-Gly	DPPH	IC_50_: 0.283 mg/mL	463.41 Da	[149]
				Superoxide radical	0.151 mg/mL		
				ABTS	IC_50_: 0.210 mg/mL		
	Pepsin	Frame	Glu-Ser-Thr-Val-Pro-Glu-Arg-Thr-His-Pro-Ala-Cys-Pro-Asp-Phe-Asn	DPPH	IC_50_: 41.37 µM	1801.00 Da	[156]
				Hydroxyl radical	IC_50_: 17.77 µM		
				Peroxyl radical	IC_50_: 18.99 µM		
				Superoxide radical	IC_50_: 172.10 µM		
Japanese flounder (*Palatichtys olivaceus*)	Pepsin	Skin	Gly-Gly-Phe-Asp-Met-Gly	In vitro, macromolecules damage	-	582.00 Da	[147]
Speckled shrimp (*Metapenaeus monoceros*)	Crude protease from *Bacillus cereus*	Shells	Protein hydrolysates	DPPH, reducing power, β-carotene	-	-	[157]
Spotless smoothhound (*Mustelus griseus*)	Trypsin	Cartilage	Gly-Ala-Glu-Arg-Pro	DPPH	EC_50_: 3.73, mg/mL	528.57 Da	[158]
				Hydroxyl radical	EC_50_: 0.25 mg/mL		
				ABTS	EC_50_: 0.10 mg/mL		
				Superoxide radical	EC_50_: 0.09 mg/mL		
			Gly-GluArg-Glu-Ala-Asn-Val-Met	DPPH	EC_50_: 1.87 mg/mL	905.00 Da	[158]
				Hydroxyl radical	EC_50_: 0.34 mg/mL		
				ABTS	EC_50_: 0.05 mg/mL		
				Superoxide radical	EC_50_: 0.33 mg/mL		
			Ala-Glu-Val-Gly	DPPH	EC_50_: 2.30 mg/mL	374.40 Da	[158]
				Hydroxyl radical	EC_50_: 0.06 mg/mL		
				ABTS	EC_50_: 0.07 mg/mL		
				Superoxide radical	EC_50_: 0.18 mg/mL		
Horse mackerel (*Magalaspis cordyla*)	Combination (pepsin, trypsin and α-chymotrypsin)	Viscera	Ala–Cys–Phe–Leu	DPPH	89.2% (treatment at 0.2 mg/mL)	518.50 Da	[159]
			-	Hydroxyl radical	59.1% (treatment at 0.2 mg/mL)	-	
Bigeye snapper (*Priacanthus macracanthus*)	Alcalase, neutrase, pyloric caeca extract	Skin	-	DPPH, ABTS, FRAP	-	-	[160]
Brownstripe red snapper (*Lutjanus vitta*)	Pyloric caeca extract	Skin	-	DPPH, ABTS, FRAP	-	-	[161]
Yellowfin sole (*Limanda aspera*)	Pepsin	Frame	Arg-Pro-Asp-Phe-Asp-Leu-Glu-Pro-Pro-Tyr	Linoleic acid model	-	13.00 kDa	[162]
Tuna	Pepsin	Backbone	Val-Lys-Ala-Gly-Phe-Ala-Trp-Thr-Ala-Asn-Gln-Gln-Leu-Ser	DPPH, hydroxyl and superoxide	-	1519.00 Da	[163]
Yellowtail fish (*Seriola lalandi*)	Protease	Scales and bone	Hydrolysates	DPPH, ABTS, reducing power, and Cu^2+^ and Fe^2+^ chelating activity	-	-	[164]
Horned turban sea snail (*Turbo cornutus*)	Protamex	Viscera	Thr-Asp-Ala	H_2_O_2_ radical, MPO inhibition	IC_50_: 646.0 ± 45.0 µM	-	[165]
			Phe-Ala-Pro-Gln-Tyr	H_2_O_2_ radical	IC_50_: 57.1 ± 17.7 µM	-	[165]
Mackerel	Alcalase	Waste	-	-	-	-	[166]
Atlantic horse mackerel (*Trachurus trachurus*)	Alcalase	Head, Skin, and Bone, Waste meat	-	DPPH, reducing power and Cu^2+^ chelating activity	-	-	[167]

### 3.3. Skin Moisturizing Effect

Marine processing by-products have been used and studied as potent alternatives to the skins of terrestrial-based animals for extract collagens because of their high availabilities, high yields, biological functions, low risk of disease transmission, and religious barriers [149]. Chitooligosaccharides, which contain N-acetyl-d-glucosamine in their structure, have been demonstrated to possess skin moisturizing potential [152]. In addition, skin moisturizing of the carboxymethyl chitosan with different MW has been demonstrated. Pig skin treated with 0.5% carboxymethyl chitosan showed a higher degree of skin moisturizing than untreated skin [168]. Compared with the lower MW carboxymethyl chitosan, higher MW showed better skin moisturizing effects, which might be related to electrical charges. The high MW carboxymethyl chitosan also decreased the loss of water.

Collagens possess potent water absorption activity, making them suitable components in food and cosmetic products. Gelatin, denatured forms of collagen and collagen peptides have been used in cosmeceuticals industries [28]. Several in vivo studies have demonstrated skin condition improvement by oral intake of fish collagen peptides. The administration of low MW collagen peptides derived from fish scales for four months has been demonstrated in moisturize hairless mice. The low MW collagen peptides were able to induce hyaluronic acid synthesis in HaCaT cells, which were mediated by induction of hyaluronic acid synthase 2 (HAS2) gene expression and downregulations of hyaluronidase 1 (HYAL1) gene expression [169]. Intake of fish collagen hydrolysates has also been demonstrated to affect dermal skin elasticity and functions [170]. Intake of fish collagen hydrolysate significantly decreased transepidermal water loss and epidermal thickness in mice. Asserin et al. [29] reported a placebo-controlled clinical trial of fish collagen peptide oral intake. Daily oral supplementation with fish collagen peptides for eight weeks significantly increased skin hydration. In addition, collagen density in the dermis increased, and the fragmentation of the dermal collagen network decreased as compared with the controls. The results also showed that fish collagen intake over eight weeks increased the skin moisture by up to 12%. Fish collagen peptide has been demonstrated to increase stratum corneal moisture content, which is an indicator of the health of this skin layer [171]. In addition, collagen peptide consumption also improves the skin viscoelasticity. Furthermore, the oral intake of collagen peptides has been tested by several studies and has proven to be safe [172]. Collagen peptides that have low MW offer many beneficial effects to beautify human skin and, therefore, offer great potential to be applied in cosmeceutical products.

## 4. Future Prospects and Challenges of Marine By-Products in the Cosmetic Industry

According to FAO data (2020), global fisheries and aquaculture production increased by more than 76 million tons between 1986 and 2018, where marine production dominates more than 64% of the worldwide output [173]. This massive production growth is attributable to breakthroughs in fishing technology and rapid innovations in aquaculture, which are substantially contributed to global fish production [174]. More than 70% of the overall production is further processed before it reaches the market, and approximately 25% of the total weight of annual marine production is discarded as by-product waste. Based on the species variety, the by-product waste may include low-quality entire fish, fish bone, fish head and tail, fish organs, skin, and fillet trims, as well as the shell of shellfish and crustaceans. Organic compounds derived from by-products have been identified as valuable resources with substantial potential for developing products with added value (such as cosmeceuticals) and resolving some of the concerns connected with environmental contamination.

Recently, marine by-product resources (e.g., by-products generated from marine resource processing) have been studied extensively because of their valuable compound and benefit to human health. Lately, several studies have been conducted to explore the utilization of marine by-products derived from cosmetics (Table 2).

The utilization of marine by-products requires good management in terms of separating, classifying, stabilizing, and preservation to ensure raw materials are in good quality and suitable conditions for delivery to industrial processing facilities. Furthermore, raw materials availability, seasonality, and accumulation analysis are some of the additional aspects that play a role in determining whether or not the by-product valorization procedures are feasible [175]. Only a few numbers of cosmetic ingredients derived from marine by-products have been put on the market and sold in significant numbers despite years of extensive study and development (Table 3). The leading causes of this are presumably exaggeration of market potential, insufficient availability of high-quality raw material for production frequently, high cost of isolating particular compounds, which are commonly produced in small quantities, and the presence of cheaper and reliable production methods such as chemical synthesis or the use of genetically modified microbes [176].

**Table 2 marinedrugs-20-00734-t002:** Novel compounds from marine by-products for cosmeceuticals.

By-Product Source	Functional Product	Processing Method	Cosmeceutical Function	Ref.
Salmon and Codfish skins	Collagen	Acid-soluble collagen (ASC) extraction	Good moisture absorption, prevents skin dehydration without irritation	[14,177]
Milkfish scale	Hydrolyzed collagen	Pepsin hydrolysis	Moisturizers, antiaging agents, and skin-whitening agents	[145]
Salmon skin	Collagen peptides	Water, protease	Wound healing	[31]
Salmon skin	Hydrolysates gelatin	Hot water, alkaline protease	Antiaging against the UV-induced photo-aging	[178]
Fish scale	Collagen peptide	Hot water, enzymatic	Improving skin elasticity	[27]
Olive flounder and Alaska pollock skins	Fish skin hydrolysates	Enzymatic hydrolysis (pepsin, alcalase, protemax)	Minimize ROS levels, enhanced the viability of UV-B irradiated HaCat cells and human dermal fibroblast	[179]
Pacific whiting skin	Hydrolysates gelatin	Hot water	Anti-photoaging, delayed skin wrinkling	[180]
Manhaden fish oil	Rich in omega-3	N.a	Reduce the irradiation effect	[51,52]
Marbled rock cod by-product	Fish oil in capsule	Solvent extraction (hexane)	Suppressed MMP-1	[55]
Shark liver	Squalen (Semosqualene^®^)	N.a	Preventing and repairing cutaneous photoaging	[181]
Codfish bone	Hydroxyapatite-Fe_2_O_3_	Calcination 700 °C	Active sunscreen filter	[66,67]
Pacific cod skin	Hydrolysates gelatin	Alkaline protease	Anti-photoaging, delaying skin wrinkling	[182]
Fringescale sardinella bone	Hydroxyapatite, hydroxyapatite- Mn, hydroxyapatite- Fe	Calcination 900 °C	Active sunscreen filter	[68]
Salmon skin	Collagen peptide	Water, protease	Antioxidant and anti-inflammatory	[183]
Tuna skin	Hydrolyzed collagen	Static hydrothermal hydrolysis	Antiaging (inhibiting tyroanase and gelatinase) and antioxidant	[25]
Codfish skin	Collagen polypeptides	Water, pepsin and alkaline protease	Moisturizer, antioxidant	[184]
Pacific cod skin	Gelatin and polypeptides	Hot water extraction, pepsin, and alkaline protease hydrolysis	Melanogenesis inhibition	[185]
Salmon skin	Gelatin hydrolysates	Enzymatic hydrolysis	Prevent collagen loss in photoaging skin caused by UV irradiation	[178]
Shrimp shell	Chitosan oligosaccharide	Enzymatic hydrolysis	Exhibit antiaging activity	[186]
Crab shell	Chitin nanofibrils, Oligochitosan-tetracycline and erythromycin	Acid hydrolysis	Prevents skin dryness, Anti-inflammatory and antioxidant (delivery system), Antibacterial (*P.acne*)	[187,188,189]
Oyster shell	Powdered oyster shell, organic shell extract	Fine grinding, Acid for decalcination, water extract	Utilize as emulsion stabilizer for cosmetic, Improving collagen content	[116,190]
Mussel and oyster shell	Shell extract	Acid aqueous extraction	Induced the synthesis of type i and iii collagens and sulfated gags	[115,191]
Pearl oyster shell	Water-soluble matric and fraction (SE4) of nacre, Nacre extract (pearl),	Water	Increase proliferation and collagen, Promoted the differentiation, Enhanced collagen synthesis in a rat skin	[109,110,111]
Outer and inner squid skins	Collagen hydrolysates	Enzymatic hydrolysis (alcalase)	Demonstrate great water-holding capacity	[192]
Squid pens	N-(2-hydroxyl) propyl-3-trimethyl ammonium chitosan chloride (HTCC)	Glycidyl trimethyl ammonium chloride (GTMAC) synthesis	Indicate good moisture absorption and retention capacity	[193]
Squid ink	Squid ink polysaccharides	Enzymatic hydrolysis (papain)	Prevent oxidative stress in human dermal fibroblast	[194]

**Table 3 marinedrugs-20-00734-t003:** Several examples of cosmetic companies utilizing marine by-product derivatives for cosmetic ingredients.

Company	Country	By-Product Resource	Bioactive Compounds	Cosmeceutical’s Function	Ref.
Finn Canada	Canada	Salmon skin	Collagen	Improve skin condition. Treat various skin problems such as wrinkles, spots, dryness, dullness, and acne	[195]
Kenney and Ross Limited	Canada	Fish skin	Collagen	Stimulates healthy skin, nails, and hair	[196]
Copalis	France	Fish skin and bone	Collagen type I-III, elastin	Skin moisturization, anti-wrinkle, skin regeneration, enhance skin elasticity,	
Revolution fibres Ltd.	New Zealand	Fish skin	Collagen	Reduce the appearance of wrinkles and sunspots	[197]
Rousselot	France	Fisk skin and bone	Collagen peptides	Skin moisturization, enhance skin collagen density	[29]
Celergen Inc	Switzerland	Fish skin	Collagen hydrolysate	Enhance skill elasticity	[27]
Abyss	France	Fish skin	Collagen hydrolysate	Reduce the appearance of wrinkles	[30]
Nuwen	France	Fish skin	Collagen hydrolysate	Skin moisturization	[198]
One Ocean	United States	Fish skin	Collagen	Skin moisturization, anti-wrinkle	[199]
Osteralia	France	Nacre	Oyster shell	Antiaging, skin nourishment	[200]

As regards the high annual production of marine by-products and only a few numbers of their added value product in the market, there is still a gap and great opportunity to develop added value products of marine by-products in the future. Green technology should be considered for marine by-product valorization because of its advantages in preserving and improving the quality and extraction efficiency and limiting functional property losses of the bioactive chemicals extracted from marine processing by-products.

The economic feasibility and technical issues of recovering cosmetic bioactive from by-products should be considered, where the circular economy concept with integrated biorefinery is a promising approach to optimize biomass use and reliability for business purposes [10]. The recovery of cosmetical active compounds from marine by-products might face issues of effective extraction and, to some extent, the marine by-product content high amount of non-cosmetical compounds that can be used for other purposes or contain multiple types of active compounds. For instance, the mollusk shell contains nacres and a large amount of calcium carbonate, so after scratching the nacre layer, the remaining part, mostly calcium carbonate, could be used for concrete biomaterials. In addition, the integrated biorefinery concept has been proposed to recover astaxanthin and chitin from crustacean shells [201]. The fish bone also contains hydroxyapatite and collagen. Therefore, the market demand and economic feasibility should be considered when choosing which type of active compounds should be recovered.

Considering the continuity of raw material is also important in choosing sources of cosmetic bioactives. For example, the bivalve processing mussel also generates byssus threads that contain collagen [202], but recovery collagen from that animal thread may face issues of material continuity, since it is available in a small amount. That continuity is critical in the formulation of cosmetic products [203]. However, increasing the sensitivity of consumers to a cosmetic product should consider using marine by-products for cosmetics where the hygienic, ecological, allergen, safety, and ethics of the cosmetic product should be compiled to be widely accepted [204].

## 5. Conclusions

Marine processing by-products have increasingly been recognized and studied in recent years as potential cosmeceutical agents. In addition, many marine processing by-products have been proven to possess potential cosmeceutical properties (e.g., skin-whitening, antiaging, skin rejuvenation properties, and moisturizing effects). However, several factors should be taken into account to develop marine by-products in cosmeceuticals. These include the mechanism of action of marine processing by-products, proper in vivo or in vitro models for testing cosmeceuticals products, hygiene, safety, and economic feasibility. There are many marine processing by-products with potential cosmeceutical properties, and only a few numbers of cosmetic ingredients derived from marine by-products have been put on the market and sold in significant numbers. Therefore, it provides challenges and opportunities for researchers to develop novel and high-value cosmeceuticals derived from marine processing by-products that can promote sustainable blue cosmetics.

## Figures and Tables

**Figure 1 marinedrugs-20-00734-f001:**
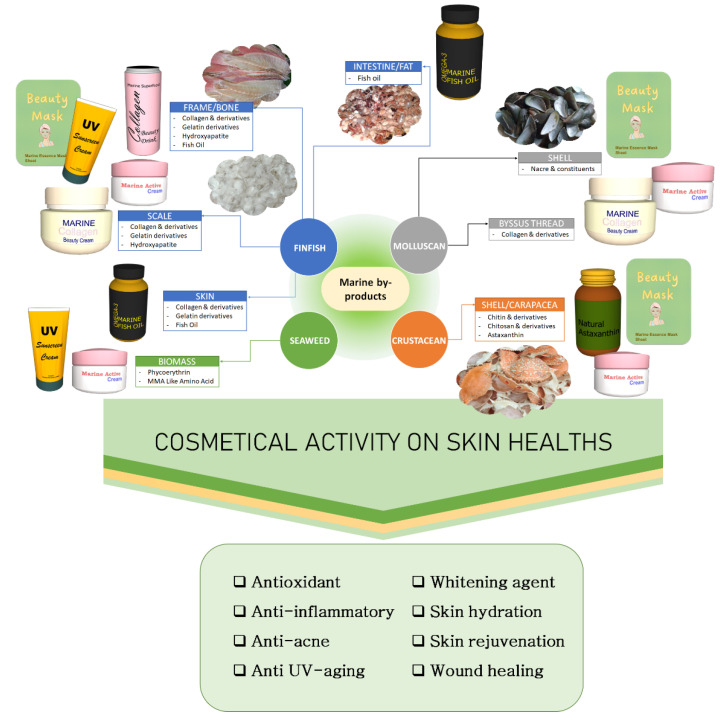
Potential utilization of marine by-products in cosmetics.

**Figure 2 marinedrugs-20-00734-f002:**
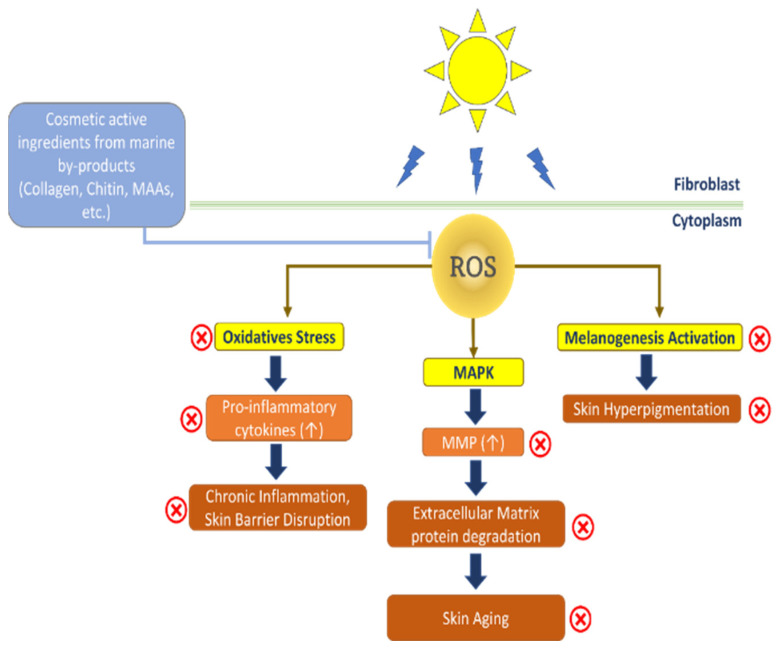
Photoprotective mechanism of cosmetic active ingredients from marine by-products on human skin.

## Data Availability

Not applicable.

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
