# Peer review of "Potential Cosmetic Active Ingredients Derived from Marine By-Products"

_marinedrugs, 2022, doi:10.3390/md20120734_

Round 1

Reviewer 1 Report

 it’s very comprehensive and well written. They have covered good amount of data on cosmetics. Only problem I noted was in the reference list. They have used issue numbers in some cases and for web sites year is missing. Other than that this is perfect. I would like to suggest the authors to include some graphics/tables etc to deliver the ideas more effectively. 

Author Response

REVIEWER 1

Comments & Suggestions for Authors:

it’s very comprehensive and well written. They have covered good amount of data on cosmetics. Only problem I noted was in the reference list. They have used issue numbers in some cases and for web sites year is missing. Other than that, this is perfect. I would like to suggest the authors to include some graphics/tables etc to deliver the ideas more effectively. 

Responses:

We agree with the reviewer and the changes have been made as suggested.

The website year and missing numbers have been corrected.

New graphic has been added.

Reviewer 2 Report

The manuscript has a satisfactory body of scientific information and is well-structured. English language errors (syntax, grammar, wording) throughout the text need correction. All acronyms should be explained (what they stand for) when they first appear in the manuscript. Also, abbreviations should be the same throughout the text, when used for the same set of words (e.g. either CaPs or CPs). Although all Tables presented, as compiled information, are very helpful to the reader, the incorporation of few (one or two) schematic/graphic representations or figures would add value to the review (e.g. chemical structures of substances, real photos of fish industry by-products vs. cosmetic end-products etc.)

Author Response

REVIEWER 2

Comments and Suggestions for Authors

The manuscript has a satisfactory body of scientific information and is well-structured. English language errors (syntax, grammar, wording) throughout the text need correction. All acronyms should be explained (what they stand for) when they first appear in the manuscript. Also, abbreviations should be the same throughout the text, when used for the same set of words (e.g. either CaPs or CPs). Although all Tables presented, as compiled information, are very helpful to the reader, the incorporation of few (one or two) schematic/graphic representations or figures would add value to the review (e.g. chemical structures of substances, real photos of fish industry by-products vs. cosmetic end-products etc.)

Response:

We agree with the reviewer comments, and the changes have been made as suggested.

All acronyms and abbreviations have been checked and revised.

The graphic has been modified and new graphic has been added.

Reviewer 3 Report

The manuscript describes the Potential Applications of Marine By-Products in Cosmeceuticals. Although this work presents an interesting topic for the scientific community, in my opinion, some substantial changes need to be done to the manuscript before considering it for publication in any journal. Please, see my comments below:

General comments:

In my opinion, the review is too long with an excessive number of references (257!). There is unnecessary information, especially in section 2 (Potential cosmeceutical ingredient from marine by-product), which does not fit with the topic of the review (at least with the title of the review).

Sometimes the authors use speculative claims, for example – in line 362: The oligomer chitosan might act as an antibacterial, anti-inflammatory, and antioxidant – without any reference. Again, in line 587: Collagen hydrolysates from marine by-products could act as TYR inhibitors. The purpose of a review is to do a comprehensive analysis of the existing literature.

I highly recommend structuring the review in another way to improve the understanding. Which is the main purpose of the review? What kind of messages do the authors want the reader to get from this review?

I have the feeling that the authors are talking in general about bioactive compounds from marine sources, without taking into consideration if it is really possible to extract them from by-products.

In general, English needs to be improved. I recommend the help of a native speaker to improve the quality and understanding of the review.

Other comments:

- The title is quite general.

- The abstract looks like an introduction.

- Figure 1 is poor quality. It is not possible to read the text. Please, replace it with a figure of high quality. Moreover, a Figure needs to be placed after its reference in the text.

- In Table 1 is necessary to include the results of the antioxidant activity for each peptide, also the assay used.

- Table 3 – are the authors sure about only 4 companies in the world using by-products for cosmeceuticals?

- The ending of the review is really poor. Which is the main conclusion of it?

- The bibliography section needs to be checked. There are a lot of mistakes. For example, the scientific names of species are italicized (e.g. ref 165, ref 176, ref 181, etc.). If the authors want to have websites as references, they need to do it appropriately.

Author Response

REVIEWER 3

Comments & Suggestions for Authors:

The manuscript describes the Potential Applications of Marine By-Products in Cosmeceuticals. Although this work presents an interesting topic for the scientific community, in my opinion, some substantial changes need to be done to the manuscript before considering it for publication in any journal. Please, see my comments below:

General comments

Comments:

-In my opinion, the review is too long with an excessive number of references (257!). There is unnecessary information, especially in section 2 (Potential cosmeceutical ingredient from marine by-product), which does not fit with the topic of the review (at least with the title of the review).

Response:

We agree with the reviewer criticsm and we have reduced the unnecessary information and references.

Comments:

-Sometimes the authors use speculative claims, for example – in line 362: The oligomer chitosan might act as an antibacterial, anti-inflammatory, and antioxidant – without any reference. Again, in line 587: Collagen hydrolysates from marine by-products could act as TYR inhibitors. The purpose of a review is to do a comprehensive analysis of the existing literature.

Response:

We have added the references.

Comments:

-I highly recommend structuring the review in another way to improve the understanding. Which is the main purpose of the review? What kind of messages do the authors want the reader to get from this review?

Response:

The manuscript has been restructuring.

More explanations of the purpose of this work have been added to improve the understanding of this work.

Comments:

-I have the feeling that the authors are talking in general about bioactive compounds from marine sources, without taking into consideration if it is really possible to extract them from by-products.

Response:

We have added our perspectives about the feasibility of the cosmeceutical development from marine by-products (part 4).

Comments:

In general, English needs to be improved. I recommend the help of a native speaker to improve the quality and understanding of the review.

Response:

The English have been checked and revised.

Other comments

Comments:

- The title is quite general.

Response:

The title has been modified.

Comments:

- The abstract looks like an introduction

Response:

We have modified the abstract.

Comments:

- Figure 1 is poor quality. It is not possible to read the text. Please, replace it with a figure of high quality. Moreover, a Figure needs to be placed after its reference in the text.

Response:

 The figure has been corrected.

Comments        :

- In Table 1 is necessary to include the results of the antioxidant activity for each peptide, also the assay used.

Response          :

The results and assays have been added in the Table.

Comments :

- Table 3 – are the authors sure about only 4 companies in the world using by-products for cosmeceuticals?

Response:

We realized that not only 4 companies mention in Table 3 using marine by-products for cosmeceuticals; therefore, Table 3 title have been modified into by-products for cosmeceuticals? Table 3. Several examples of cosmetic company utilized marine by-product derivatives for cosmetic ingredient

Comments:

- The ending of the review is poor. Which is the main conclusion of it?

Response:

Conclusions have been added

Comments:

- The bibliography section needs to be checked. There are a lot of mistakes. For example, the scientific names of species are italicized (e.g. ref 165, ref 176, ref 181, etc.). If the authors want to have websites as references, they need to do it appropriately.

Response:

The bibliography has been corrected.

Round 2

Reviewer 3 Report

The author’s replies are general and not specific enough to check what they changed. Usually, you specify the line number when you answer a question or use the track changes function.

-I still think that the review is too long with unnecessary information and with a high number of references. For example, do the authors think that the information in lines 225-235 is really necessary?

- The authors said that they reduced the unnecessary information, but I could not check which text parts were removed because the authors did not use the track changes function. Please, use the track changes function to see the parts which have been removed.

-The quality of Figure 1 is still low quality. I think it is because it is a copy and paste from other sources. Have the authors the right to use them?

- Table 3. The authors said: “We realized that not only 4 companies mention in Table 3…”. Then, why did the authors not include the other companies?

- The English needs to be improved. The text is really difficult to read. There are a lot of abbreviations, which sometimes are not explained. Again, it is not possible to check the changes that the authors did. Actually, there are still a lot of mistakes, for example:

§  Docosahexanoic acid (DHA) and Eicosapentanoic acid (EPA) are written with capital letters…

§  Synthetic and natural CPs… (line 245) Do the authors want to say CaPs? The abbreviation CPs appears 5 times without any explanation.

§  pro-Vitamin A is written with capital letters (line 389), same melatonin-Vitamin E-Betaglucan ( lines 330) or Metallo Proteinase I (336)

§  In vivo and in vitro need to be in italics

§  Explanation of the abbreviation EMB is not in the text.

§  There are mistakes when the authors are citing other works, for example, Ahmadkelayeh, et al. / Vicente, et al. / Morganti, et al. - Why there is a comma between the author name and et?

§  The degree symbol is written every time in a different way.

§  Mycosporine-like amino acids appear in the text before the abbreviation.

§  Why “anti-aging, skin rejuvenation properties” are written in italics? (line 746)

Author Response

18 November, 2022

For the kind attention

Reviewer of Marine Drugs,

Dear Sir,

We would like to thank you the reviewer for your thoughtful comments and valuable suggestions, which helped us in improving the quality of the manuscript.  We have carefully considered the comments the editor and reviewers provided, and we have revised the manuscript accordingly. We greatly appreciate the efforts of the editor and reviewers and believe our revisions have generated an improved product. Below are the responses (red color) for all the comments. The line numbers included in this response letter refer to those in the revised manuscript. Revised versions of 1st and 2nd rounds in the manuscript and related reports on peer review are given in blue color for your convenience.

Sincerely yours,

Evi
